# Masked Siamese ConvNets: Towards an Effective Masking Strategy for General-purpose Siamese Networks

## Abstract

Siamese Networks are a popular self-supervised learning framework that learns useful representation without human supervision by encouraging representations to be invariant to distortions. Existing methods heavily rely on hand-crafted augmentations, which are not easy to adapt to new domains. To explore a domain-agnostic siamese network, we investigate using masking as augmentations in siamese networks. Recently, masking for siamese networks has only been shown useful with transformer architectures, e.g. MSN (Assran et al., 2022) and data2vec (Baevski et al., 2022). In this work, we identify the underlying problems of masking for siamese networks with arbitrary backbones, including ConvNets. We propose an effective masking strategy and demonstrate its effectiveness on various siamese network frameworks. Our method generally improves siamese networks' performances in the few-shot image classification, and object detection tasks.

## 1 Introduction

Self-supervised learning aims to learn useful representations from scalable unlabeled data without relying on human annotation. It has widely used in natural language processing (Devlin et al., 2019; Zhang et al., 2022; Brown et al., 2020), speech recognition (van den Oord et al., 2018; Hsu et al., 2021; Schneider et al., 2019; Baevski et al., 2020) and other domains (Rives et al., 2021; Rong et al., 2020). Recently, self-supervised visual representation learning has also become an active research area.

First introduced by Bromley et al. (1993), the siamese network (Chen et al., 2020a;b; He et al., 2020; Chen et al., 20220; 2021; Caron et al., 2020; Grill et al., 2020; Chen & He, 2020; Wang et al., 2022; Zbontar et al., 2021; Bardes et al., 2021) is one promising approach among many self-supervised learning approaches and outperforms supervised counterparts in many visual benchmarks. It encourages the encoder to be invariant to human-designed augmentations, capturing only the essential features. Practically, in the vision domain, the siamese network methods rely on domain-specific augmentations, such as cropping, color jittering and Gaussian blur, which do not transfer well to other domains. Therefore, it is desired to have a general augmentation approach for siamese networks that require minimal domain knowledge and can generalize.

Among various augmentations, masking the input is one of the simplest and effective choices, which has been demonstrated to be useful for language (Devlin et al., 2019) and speech (Hsu et al., 2021). However, not until the recent success of vision transformers (ViTs) (Dosovitskiy et al., 2021; Touvron et al., 2021) can vision models leverage masking as a general augmentation. Self-supervised learning with masked inputs has demonstrated more scalable properties when combined with ViTs (He et al., 2021; Bao et al., 2021; Zhou et al., 2021; Baevski et al., 2022). Unfortunately, siamese networks with naive masking do not work well with arbitrary architecture, e.g., ConvNets (He et al., 2016; Liu et al., 2022).

We identify the underlying issues behind masked siamese networks with ConvNets. We argue that ConvNets do not have a mechanism to encode null information. In addition, masking will introduce parasitic edges. We propose a preprocessing procedure to solve these problems. We present several general-purpose designs that allow siamese networks to benefit from masked inputs. Experiments

show that siamese networks with ConvNets backbone can benefit from masked inputs with our masking strategy.

We summarize our contributions below:

- We discuss the role of augmentations in siamese networks and explore a general-purpose augmentation approach of masking. We identify the underlying problem of masking for siamese with ConvNets backbones.

- We propose a preprocessing step to overcome the problem behind masked siamese networks with ConvNets. We present a series of masking designs which allow masking to benefit the siamese network with ConvNets backbones.

- We propose Masked Siamese ConvNets (MSCN), an effective masking strategy for general-purpose siamese networks. Our method can be applied to various siamese network frameworks, and it demonstrates competitive performances on few-shot image classification benchmarks and outperforms previous methods on object detection benchmarks.

## 2 RELATED WORKS

### 2.1 SIAMESE NETWORKS

Self-supervised visual representation learning has become an active research area since they have shown superior performances over supervised counterparts in recent years. One promising approach is to learn useful representations by encouraging them to be invariant to augmentations, known as siamese networks or joint-embedding methods (Misra & van der Maaten, 2020; Chen et al., 2020a;b; He et al., 2020; Chen et al., 20220; 2021; Caron et al., 2020; Grill et al., 2020; Chen & He, 2020; Wang et al., 2022; Zbontar et al., 2021; Bardes et al., 2021). These methods use different mechanisms to prevent collapse, and they all rely on carefully designed augmentations such as random resized cropping, color jittering, grayscale and Gaussian blur. These augmentations prevent the encoder from only using trivial features. Empirically, siamese networks with these standard augmentation settings usually work well with arbitrary architectures, including ResNets (He et al., 2016) and ViTs (Dosovitskiy et al., 2021). Their representations are label-efficient (Assran et al., 2021; 2022), more robust (Hendrycks et al., 2019), and have improved fairness (Goyal et al., 2022). In addition, Siamese networks have been demonstrated to benefit from scalable data (Goyal et al., 2021a).

### 2.2 REPRESENTATION LEARNING WITH MASKED INPUTS

Masking the input is one of the simplest methods to corrupt the information input and could be applied to a wide range of data types. It has been mostly used in two scenarios.

The first is for denoising autoencoder frameworks (Vincent et al., 2010; 2008). Motivated by its success in NLP (Devlin et al., 2019; Brown et al., 2020) with transformers (Vaswani et al., 2017), various visual representation learning methods (He et al., 2021; Bao et al., 2021; Zhou et al., 2021) using ViTs have also shown benefit from masked inputs. These methods have proven to be a promising general-purpose self-supervised learning approach.

The second is for siamese networks. Siamese networks can benefit from masked inputs (Baevski et al., 2022; Assran et al., 2021) where masking serves as an extra augmentation. These methods are able to learn more transferable representations and have the benefit of being label-efficient. These works are limited to ViT architectures. However, no previous work has shown that the masking approach can work equally well with arbitrary ConvNets.

## 3 AUGMENTATIONS FOR SIAMESE NETWORKS

In this section, we discuss the role of augmentations in siamese networks and outline several design principles which will be used to guide our masking strategy.

### 3.1 PRELIMINARIES

Siamese networks for visual representation learning start with creating two random views $\mathbf{x}_1$ and $\mathbf{x}_2$ from the same input $\mathbf{x}$, with two different sets of random augmentation transformations $T_\phi$ and $T_{\phi'}$ to each view. These methods then train an encoder $f_\theta(\cdot)$ and a projector $q_\theta(\cdot)$ to minimize $L_p = \mathbb{E}_{\phi,\phi'} \|q_\theta(f_\theta(T_\phi(\mathbf{x}_1))) - q_\theta(f_\theta(T_{\phi'}(\mathbf{x}_2)))\|^2$, known as the positive term of the siamese network loss function. To simplify the notation, we will ignore the projector $q_\theta(\cdot)$ in all of our following discussions, since we could always consider the projector as a part of the encoder.

However, if the encoder $f_\theta(\cdot)$ is trained with the positive term alone, it $f_\theta(\cdot)$ will quickly converge to a collapsed solution that produces a constant representation for all inputs. Preventing collapse can be solved by various frameworks, including contrastive loss Chen et al. (2020a); He et al. (2020), redundancy reduction Zbontar et al. (2021); Bardes et al. (2021), clustering Caron et al. (2020) and distillation Grill et al. (2020); Chen & He (2020); Caron et al. (2021). These methods explicitly or implicitly prevent $\mathbb{E}_{\phi,\phi'}[\|f_\theta(T_\phi(\mathbf{x})) - f_\theta(T_{\phi'}(\mathbf{x}'))\|^2]$ becoming too small for $\mathbf{x}, \mathbf{x}'$ coming from different images. This is known as the negative term of the siamese networks loss function. Although the negative term design is an important topic of siamese network methods, it is relatively independent of augmentation choice.

In this work, we focus on the positive term to analyize the role of augmentations $T_\phi$. Data augmentation plays an important role in guiding encoders $f_\theta(\cdot)$ to learn useful representations of the input so that they can be used for downstream tasks. With weak augmentation, we could find a encoder $g$ based on simple input statistics such that $\|g(T_\phi(\mathbf{x})) - g(T_{\phi'}(\mathbf{x}))\|$ is small for all $\phi$ and $\phi'$, then the encoder $f$ only needs to capture those superficial features to minimize the positive term of the loss function. With aggressive augmentations, it becomes difficult to find a encoder $g$ such that $\|g(T_\phi(\mathbf{x})) - g(T_{\phi'}(\mathbf{x}))\|$ is small for all $\phi$ and $\phi'$ due to the mutual information between two augmented views is destroyed. Then the encoders $f_\theta(\cdot)$ are impossible to capture that information which is potentially useful for downstream tasks.

Furthermore, other than assessing the augmentations based on how aggressive it is, Huang et al. (2021) shows that the generalization ability of contrastive self-supervised learning depends on the concentration of augmented data within the same latent class. Intuitively, it means that to increase the performance of downstream tasks that require the semantic information of the data, the augmentations we applied to input data should be able to increase the probability of similar augmented views of images from the same latent class, or decrease the probability of similar augmented views of images from the different latent classes.

### 3.2 AUGMENTATION DESIGNING PRINCIPLE

Following the discussion in 3.1, we present three design principles of data augmentations for siamese networks:

1. Prevent easy solutions such that encoder $f_\theta(\cdot)$ only captures certain superficial features;

2. Keep as much mutual information between the two views as possible;

3. Increase the probability of similar augmented views of images from the same latent class, or decrease the probability of similar augmented views of images from the different latent classes.

## 4 PROBLEMS IN MASKED SIAMESE NETWORKS WITH CONVNETS

Siamese networks using masking as an extra augmentation have demonstrated competitive performances (Assran et al., 2022; Baevski et al., 2022) with ViT backbones. Naively replacing ViTs with ConvNets results in significantly worse performances. Here, we identify its underlying problems. See Figure 1.

**Masking As Missing Information** - The main reason masking as augmentation performs poorly on siamese networks with ConvNets is that masking represents missing information, and there is no good choice of assigned values for continuous input. Figure 1b shows the color histograms for the original input and its masked version. Naively assigning zero or noise or even a trainable value will

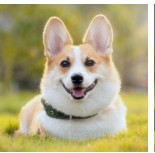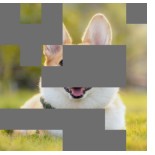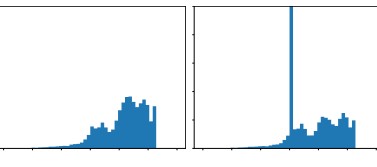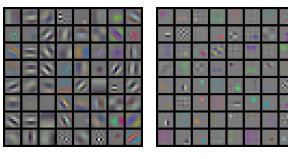

(a) input images        (b) histogram        (c) learned convolution kernels

Figure 1: Underlying problems in masked siamese networks with ConvNets. Each pair in a subfigure corresponds to the original input and the masked version. (a) visualization of the input image and the masked version; (b) the color histogram was distorted with masks. The values close to the assigned mask value now represent missing information and could confuse the network; (c) visualization of the encoder's first convolutional layer kernels after pretraining.

confuse the network. For ViTs, this problem is automatically avoided by the self-attention mechanism that does not attend to the masked area. Hence, it is essential for the network to identify masking as missing information with proper preprocessing.

**Masking Introduces Parasitic Edges** - The convolutional kernels are well known for their edge detection behaviors (LeCun et al., 1998). Applying masks creates a large number of parasitic edges in the images. The feature maps generated by edge-detecting kernels are drastically distorted; hence, these kernels are suppressed during training by siamese networks. More severally, these parasitic edges will remain in the intermediate feature maps and affect all the convolutional layers. In Figure 1c, we visualize the encoder's first convolutional layer kernels pretrained with standard augmentation or masked input. Due to the parasitic edges, many kernels collapsed to trivial blank features. ViTs dodge this problem by simply matching the patch boundaries to the mask.

## 5 METHODS

### 5.1 PREPROCESSING

To solve the problems caused by masking in siamese networks with ConvNets backbones as discussed in Section 4, we propose to apply a high-pass filter during the preprocessing stage. See Figure 2.

*High-pass filter allows zeros to represent null information* - With an high-pass filter, the zero values in the input image now represent null information instead of a regular pixel. Therefore, by applying an extra mask with zero value perfectly fit the value distribution and results in minimal information distortion change to the regular pixels.

*High-pass filter elimiates parasitic edges* - We observe that the masking edges on the high-pass filtered image becomes invisible. See Figure 2. Empirical results also verify that the ConvNets encoder is able to learn useful edge detection kernels.

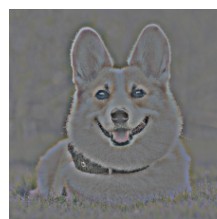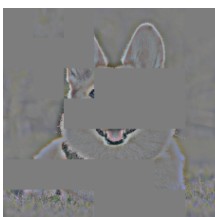

Figure 2: **Preprocessing**: A high-pass filter is able to solve the problem caused by masks on ConvNets. The high-pass filter shifts the pixel value distribution so that zeros represent null information. These zeros mean that the original information can be interpolated from surrounding pixels. In addition, the mask edges on the image become invisible after applying the high-pass filter.

We quantitatively show that a high-pass filter as an extra preprocessing step significantly improves siamese networks' performance with masked inputs. We start with applying two random grid masks with grid size 32 on the same random crop with a fixed 30% masking ratio. We do not apply any other

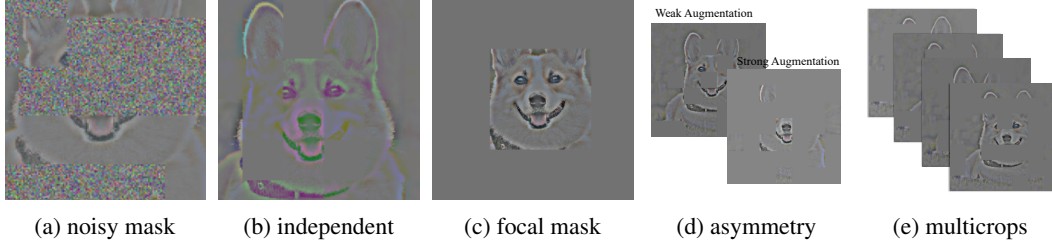

| (a) noisy mask | (b) independent | (c) focal mask | (d) asymmetry | (e) multicrops |

Figure 3: **Masking Strategy**: We present five effective design components that allow siamese networks to benefit from masked input. These design components are general-purpose and can work with various input formats.

augmentations, such as random resized crops or color distortions. This mask-only setting achieves a non-trivial 21.0% linear probe accuracy on ImageNet. By adding a high-pass filter, the accuracy improves to 30.2%. We empirically find that a smaller high-pass filter kernel $\sigma = 5$ is optimal, see Appendix B.

One may suspect that a high-pass filter will eliminate low-frequency information and results in learning worse representations. We empirically show that a high-pass filter will cause negligible change in terms of performance. We conduct supervised training experiments with or without a high-pass filter and evaluate their image classification accuracy. A ResNet-50 with a high-pass filter reaches 76.2% ImageNet supervised Top-1 accuracy, while the one without a high-pass filter gets 76.0%. These two numbers are within an error range.

## 5.2 MASKING STRATEGY

Besides standard augmentations, adding masks could prevent the encoder $f_\theta(\cdot)$ from learning some superficial features, as discussed in Sec 5, but at the same time, masking might also introduce new superficial features that confuse the encoder. We propose several masking designs to prevent such behaviors.

**Noisy Mask** - Inspired by masked language modeling Devlin et al. (2019) where masked tokens were replaced by random tokens with a certain probability, we find that adding Gaussian noise to the masked area is beneficial and reduces the siamese networks' reliance on augmentations such as color jittering. A noisy mask prevents the network from easily overfitting to use the color histogram as the entire feature, which is similar to the role of applying color-jittering on the unmasked area (Chen et al., 2020a).

**Channel-Independent Mask** - In a discrete space, it is natural to mask the entire patch or token. However, for continuous input with ConvNets backbone, it is essential for the network to attend to not only spatial correlation but local channel-wise dependency. In standard siamese networks, this is done by handcrafting augmentations such as color-jittering and grayscale. Here, we propose to add such functionality for masks by introducing a channel-independent mask. In addition to standard spatial-wise masking, where we apply the same mask on three color channels, we generate three random masks and apply them to each color channel separately. Thus, this design is agnostic to input format as it blurs the difference between spatial and channel dimensions. We suspect that this design can further improve high-dimensional input such as RGBD images or videos.

**Combining Random Masks and Focal Masks** - In siamese networks with ViTs, Assran et al. (2022) show that a mixture of random grid masks and focal masks can significantly boost the performance of the siamese network. This is also similar to (Bao et al., 2021) where larger blockwise masks are combined with randomly smaller ones. Compared to a random grid mask, a focal mask decreases the probability of similar augmented views of images from the different latent classes. Intuitively, the larger a patch is, the smaller the probability that images from the different latent classes could both generate the patch. Therefore, the siamese network will be encouraged to learn semantic-level features. In our design, we randomly apply focal masks or distributed grid masks for each view.

**Asymmetric Augmentation** - Wang et al. (2022) highlighted the importance of asymmetry for siamese networks in augmentations on various siamese network frameworks. Here, we find that

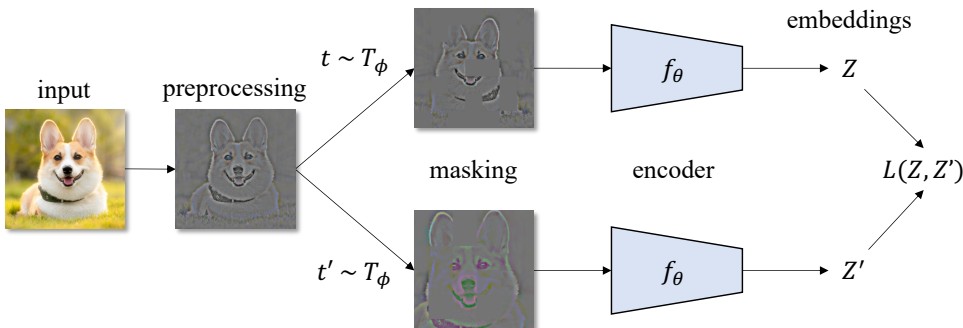

Figure 4: **Masked Siamese ConvNets (MSCN) framework**. MSCN first generates multiple views from the preprocessed image using a series of standard augmentations. Then it applies random masks on each view using our masking strategy. An ConvNet encoder computes the representations of these masked views. Then it applies joint-embedding loss is applied to pairs of these representation vectors.

masking asymmetrically between the two branches in the siamese networks improves learned representations. This has also been used in masked siamese networks with ViTs (Assran et al., 2022) that one branch always takes an unmasked view. Empirically, we find that applying masking on both branches is essential for good representations, especially for frameworks that do not use momentum encoders (He et al., 2020; Grill et al., 2020). In our design, we will apply masking alongside other augmentations on both branches with different magnitudes.

**Multicrops** - Besides the two main views used in siamese networks, Multicrops (Caron et al., 2020) feed additional views for each image and apply siamese networks between some pairs of them. Multicrops have been proven beneficial in various frameworks. Instead of relying on small resolution views used in previous methods (Caron et al., 2020; Assran et al., 2022), we simply apply multiple views with the same resolution and apply siamese network loss on every pair of them. Specifically, we propose to use two extra masked views. Thus, our strategy does not assume input format.

### 5.3 MASKED SIAMESE CONVNETS FRAMEWORK

The overall Masked Siamese ConvNets (MSCN) architecture is shown in Figure 4. This framework can be a simple add-on to various siamese network frameworks with arbitrary backbone architecture. During preprocessing, the high-pass filter is conducted after normalization. During the augmentation stage on each branch, the masking is applied after other augmentations.

## 6 RESULTS

In this section, we evaluate the representations obtained by a Masked Siamese ConvNets pretrained on the ImageNet-1K dataset (Russakovsky et al., 2015) with ResNet-50 backbone (He et al., 2016). We train MSCN with different joint-embedding losses, namely SimCLR and BYOL. The network is pretrained for 800/1000 epochs, including 10 epochs of warm-up, and uses a cosine learning rate schedule. We use the LARS optimizer (You et al., 2017) with a batch size of 4096. All the hyperparameters, including learning rate, closely follow the original SimCLR and BYOL implementation so that we could have a fair comparison. For each model, the pretraining is distributed across 32 V100 GPUs and takes approximately 180 hours. We list all the pretraining and evaluation details in supplementary material.

### 6.1 IMAGE CLASSIFICATION

We first evaluate the representations on the ImageNet-1K dataset using linear probe and semi-supervised classification. We compare MSCN with baselines in Table 1.

For linear probe, we train a linear classifier on $100\%$ of the labels with frozen weights. MSCN improves SimCLR but performs slightly worse with the BYOL baseline. We suspect BYOL may require a different asymmetric masking setting due to its asymmetric nature of the framework.

For semi-supervised classification, we finetune the network using $1\%$ of the labels. MSCN demonstrates superior performances. The advantage of masked siamese networks in few-shot image classification tasks has also been observed by Assran et al. (2022) using ViTs.

We compare the effect of masking on ConvNets and ViTs in Table 2. MSCN with a ConvNet backbone demonstrates similar behaviors to MSN with a ViT backbone.

Table 1: **Image Classification on ImageNet-1K.** We first evaluate the MSCN representations with ImageNet-1k classification. For linear evaluation, we freeze the backbone and train a linear classifier with all the labels. For low-shot semi-supervised learning, we finetune the network with $1\%$ of the labels. MSCN demonstrates improvement over baselines on both SimCLR and BYOL frameworks on both settings.

| Method | Linear Evaluation | Finetuning |
|---|---|---|
| Label fraction | $100\%$ | $1\%$ |
| SimCLR (Chen et al., 2020a) | 69.5 | 48.3 |
| +MSCN | 71.5 (+2.0) | 54.0 (+5.7) |
| BYOL (Grill et al., 2020) | 74.3 | 53.2 |
| +MSCN | 74.5 (+0.2) | 55.2 (+2.0) |

Table 2: **Effect of Masking on ConvNets and ViTs**. We compare the effect of masking on ConvNets and ViTs on ImageNet-1K with linear probe. MSCN with a ConvNet backbone demonstrates similar behaviors to MSN with a ViT backbone.

| Method | Architecture | Parameters | Use Mask | Accuracy |
|---|---|---|---|---|
| Supervised (Touvron et al., 2021) | ViT-S | 22M | | 79.9 |
| DINO (Caron et al., 2021) | ViT-S | 22M | ✗ | 78.3 |
| MSN (Assran et al., 2022) | ViT-S | 22M | ✓ | 76.9 |
| Supervised (He et al., 2016) | ResNet-50 | 24M | | 76.5 |
| BYOL (Grill et al., 2020) | ResNet-50 | 24M | ✗ | 74.3 |
| MSCN (ours) | ResNet-50 | 24M | ✓ | 74.5 |

## 6.2 TRANSFER LEARNING

We then evaluate the representations by transferring the network to other downstream tasks. We report the transferred image classification results on iNaturalist 2018 (Horn et al., 2018) dataset and Places-205 (Zhou et al., 2014) dataset in Table 3. In Table 4, we report the object detection and instance segmentation performance on VOC07+12 (Everingham et al., 2009) and COCO datasets (Lin et al., 2014).

Comparing MSCN with other methods in Table 3. We observe similar results as Table 1, that MSCN improves SimCLR but performs worse with BYOL. We still suspect that the worse performance of BYOL could come from the non-optimal masking hyperparameters. Since most of the objects in Places-205 have a larger scale than the objects in iNaturalist 2018, the non-optimal masking hyperparameters cost more harm on the smaller scale objects than on larger-scale objects.

For the object detection and instance segmentation tasks, MSCN demonstrates superior performances over previous siamese network frameworks on VOC07+12 detection task and performs comparably to the state-of-the-art representation learning methods on COCO dataset.

Table 3: Image classification transfer learning with a ResNet-50 pretrained on ImageNet-1K. We follow the standard evaluation protocol that trains the linear classifiers on fixed features with the same hyperparameters as other methods except for the learning rate.

| Method | Place-205 | iNat18 |
|---|---|---|
| SimCLR Chen et al. (2020a) | 52.5 | 37.2 |
| +MSCN | 53.8 (+1.3) | 38.2 (+1.0) |
| BYOL Grill et al. (2020) | 54.0 | 47.6 |
| +MSCN | 54.5 (+0.5) | 45.7 (-1.9) |

Table 4: Object detection and instance segmentation transfer learning with a ResNet-50 pretrained on ImageNet-1K. All VOC07+12 results using Faster R-CNN (Ren et al., 2015) with C4 backbone variant (Wu et al., 2019) finetuned 24K iterations. All COCO results using Mask R-CNN (He et al., 2017) with C4 backbone variant (Wu et al., 2019) finetuned using the 1× schedule. Methods with an asterisk are reproduced by Chen & He (2020). 0.3 within the best are underlined

| Method | VOC07+12 det | | | COCO det | | | COCO instance seg | | |
|---|---|---|---|---|---|---|---|---|---|
| | $AP_{all}$ | $AP_{50}$ | $AP_{75}$ | $AP^{bb}$ | $AP^{bb}_{50}$ | $AP^{bb}_{75}$ | $AP^{mk}$ | $AP^{mk}_{50}$ | $AP^{mk}_{75}$ |
| Supervised | 53.5 | 81.3 | 58.8 | 38.2 | 58.2 | 41.2 | 33.3 | 54.7 | 35.2 |
| MoCo v2 | 57.4 | 82.5 | 64.0 | 39.3 | 58.9 | 42.5 | 34.4 | 55.8 | 36.5 |
| SwAV | 56.1 | 82.6 | 62.7 | 38.4 | 58.6 | 41.3 | 33.8 | 55.2 | 35.9 |
| SimSiam | 57 | 82.4 | 63.7 | 39.2 | 59.3 | 42.1 | 34.4 | 56.0 | 36.7 |
| Barlow Twins | 56.8 | 82.6 | 63.4 | 39.2 | 59.0 | 42.5 | 34.3 | 56.0 | 36.5 |
| SimCLR* | 55.5 | 81.8 | 61.4 | 37.9 | 57.7 | 40.9 | 33.3 | 54.6 | 35.3 |
| BYOL* | 55.3 | 81.4 | 61.1 | 37.9 | 57.8 | 40.9 | 33.2 | 54.3 | 35.0 |
| MSCN | 57.5 | 83.0 | 64.4 | 39.1 | 59.1 | 42.1 | 34.2 | 55.7 | 36.4 |

## 6.3 ABLATION STUDY

We conduct ablation experiments to gain insights into our masking design strategy. By default, we pretrain MSCN for 100 epochs with the SimCLR framework. We measure the performance by linear probe accuracy on ImageNet-1K.

**Masking Ratio** - We first explore the optimal masking ratio in Figure 5a. A small masking ratio of 15% is optimal for a ResNet-50 backbone. This matches the observation in (Assran et al., 2022) that smaller networks (e.g. ViT-S) prefer a smaller masking ratio. We also observe that the accuracy is relatively stable against the masking ratio up to 50% with our masking strategy.

**Masking Grid Size** - Mask grid size is an important hyperparameter that controls the balance between local and global features in the input, and determine what the siamese network will learn. In masked siamese networks with ViTs, the masking grid size is fixed, and it is always set to match the patch boundaries. However, an optimal masking grid size can vary. In Figure 5b, we show that the siamese networks can benefit from a more appropriate mask grid size. We observe a large grid size of 32 is optimal for our current masking strategy with ConvNets backbone.

**Focal Mask Probability** - We explore the optimal focal mask probability in Figure 5c. Combining focal masks with random grid masks leads to significant improvements. It remains an interesting open direction on how to optimally mix local features with global ones.

**Independent Mask Probability** - We explore the optimal independent mask probability in Table 5d. Independent mask with high probability results in significant accuracy boost.

## 7 DISCUSSION AND FUTURE DIRECTIONS

**General-purpose Siamese Networks** - Human-designed augmentations leveraging domain knowledge are essential for siamese networks to learn useful representations. Similar to masked siamese

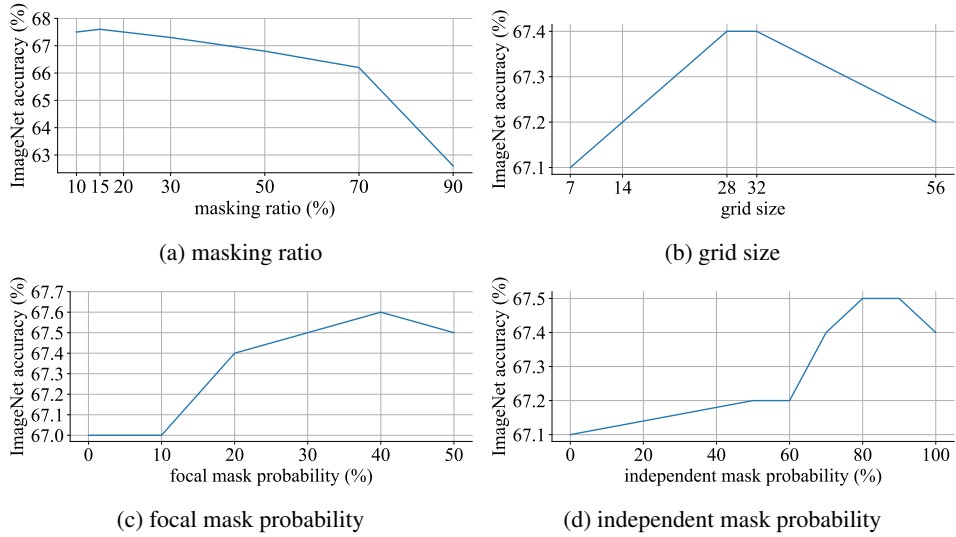

Figure 5: **Ablation Study**. (a) a masking ratio of $15\%$ is optimal for masked siamese ConvNets; (b) masked siamese ConvNets prefer a large masking grid size of 32; (c) a focal mask probability of $40\%$ is optimal; (d) independent mask with high probability benefits siamese networks.

networks with ViTs (Assran et al., 2022), our approach still combines standard augmentations with masking by proposing masking as an extra augmentation. However, it is desirable to find a domain-knowledge-free augmentation strategy so that this approach can be applied to more general domains or out-of-distribution scenarios. And it is reasonable to believe that the importance of domain-knowledge-specific augmentations will diminish with an increasing amount of pretraining data.

**Masked ConvNets for Masked Image Models** - Besides siamese networks, masked image modeling based on denoising autoencoders (Vincent et al., 2008; 2010) is another promising approach for visual representation learning. These methods (Bao et al., 2022; He et al., 2021) have demonstrated impressive performances in visual representation learning. Unfortunately, masked autoencoders also fail to work with ConvNets caused by similar problems as mentioned in Sec 4 for siamese networks. Even though the mask here serves a different purpose, we suspect that our design in Masked Siamese ConvNets may also apply to masked image modeling with ConvNets. We hope the discovery in this paper may shed light on general self-supervised learning and reduce the requirement for inductive bias of different architectures.

## 8 CONCLUSION

This work explores whether masking can be applied as an general-purpose augmentation to siamese networks with arbitrary backbones, including ConvNets. We first present the problems introduced by the use of masking as augmentation. We then propose an effective masking strategy and demonstrate its effectiveness on various siamese network frameworks. Our method performs competitively on few-shot image classification benchmarks and outperforms previous methods on object detection benchmarks.

## REPRODUCIBILITY STATEMENT

The pretraining code can be found in the supplementary material. We also provide a detailed implementation setup for pre-training and downstream experiments in Appendix A. After publication, we will provide pretrained checkpoints and open-source the code on a public repo.

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

# A    IMPLEMENTATION DETAIL

## A.1    PRETRAINING

We closely follow the original setting in (Chen et al., 2020a) for our MSCN (w/ SimCLR) pretraining and original setting in (Grill et al., 2020) for our MSCN (w/ BYOL) pretraining.

**Augmentation** - For both methods, we use the same augmentation methods. Each augmented view is generated from a random set of augmentations from the same input image. We apply a series of standard augmentations for each view, including random cropping, resizing to 224x224, random horizontal flipping, a random color-jittering, randomly converting to grayscale, and a random Gaussian blur. These augmentations are applied symmetrically on two branches.

**Architecture** - For MSCN (w/ SimCLR), the encoder is a ResNet-50 network without the final classification layer followed by a projector. The projector is a two-layer MLP with input dimension 2048, hidden dimension 2048, and output dimension 256. The projector has ReLU between the two layers and batch normalization after every layer. This 256-dimensional embedding is fed to the infoNCE loss. We use a temperature $0.2$ for the infoNCE loss.

For MSCN (w/ BYOL), the online encoder is a ResNet-50 network without the final classification layer. The online projector is a two-layer MLP with input dimension 2048, hidden dimension 4096, and output dimension 256. The predictor is a two-layer MLP with input dimension 256, hidden dimension 4096, and output dimension 256. The projector and the predictor have ReLU between the two layers and batch normalization after every layer except the final linear layer. The target encoder and projector are the exponential moving average of the online encoder and projector, with an initial momentum $\tau = 0.996$ with a cosine decay schedule to $1.0$ during the pretraining.

**Optimization** - We follow the training protocol in (Zbontar et al., 2021) . For MSCN (w/ SimCLR), we use a LARS optimizer and a base learning rate $4.8$ using cosine learning rate decay schedule. We pretrain the model for 800 epochs with 10 epochs warm-up with batch size 4096.

For MSCN (w/ BYOL), we use a LARS optimizer and a base learning rate $3.2$ using cosine learning rate decay schedule. We pretrain the model for 1000 epochs with 10 epochs warm-up with batch size 4096.

## A.2    LINEAR PROBE ON IMAGENET

We closely follow the setting used in (Zbontar et al., 2021) for our linear probe evaluation. The linear classifier is trained for 100 epochs with a base learning rate of $1.0$ and a cosine learning rate schedule. We minimize the cross-entropy loss with the SGD optimizer with momentum and weight decay of $10^{-6}$. We use a batch size of 256.

At training time, we use random resized crops to 224x224, followed by random horizontal flip and a High-pass filter. At test time, we resize the image to 256×256 and center-crop it to a size of 224×224 and followed by a High-pass filter.

## A.3    FINETUNING

We closely follow the setting used in (Zbontar et al., 2021) for our finetuning evaluation. We finetune the ResNet-50 encoder for 20 epochs with a learning rate of $0.002$ and the classifier with a base learning rate $0.5$. The learning rate is multiplied by a factor of $0.2$ after the 12th and 16th epoch. We minimize the cross-entropy loss with the SGD optimizer with momentum and do not use weight decay. We use a batch size of 256. The image augmentations are the same as in the linear evaluation setting.

## A.4    OBJECT DETECTION

We use the detectron2 library (Wu et al., 2019) for training the detection models. All the configuration files are from the VISSL library (Goyal et al., 2021b), which are closely follow the evaluation settings from (He et al., 2020). The backbone ResNet50 network for Faster R-CNN (Ren et al., 2015) and Mask R-CNN (He et al., 2017) is initialized using our pretrained model.

**VOC07+12** We use the VOC07+12 (Everingham et al., 2009) trainval set of 16K images for training a Faster R-CNN C-4 backbone for 24K iterations using a batch size of 16. The initial learning rate for the model is 0.085 which is reduced by a factor of 10 after 18K and 22K iterations. We use linear warmup (Goyal et al., 2019) with a slope of 0.333 for 1000 iterations.

**COCO** We train Mask R-CNN C-4 backbone on the COCO (Lin et al., 2014) 2017 train split for 90K iterations using a batch size of 16 The initial learning rate for the model is 0.05 which is reduced by a factor of 10 after 60K and 80K iterations. We use linear warmup (Goyal et al., 2019) with a slope of 0.333 for 1000 iterations. We report results on the 2017 val split.

# B    ADDITIONAL ABLATION STUDY

We conduct additional ablation experiments to gain insights into our masking design strategy. By default, we pretrain MSCN with SimCLR loss for 100 epochs. We measure the performance by linear probe accuracy on ImageNet-1K.

**High-pass Sigma** - We explore the optimal high-pass filter $\sigma$ in Figure 6. In addition to the varying $\sigma$ for pretraining, we also update the high-pass filter $\sigma$ for the transformation during evaluation. In practice, we prefer a small $\sigma$ because there is less computational overhead.

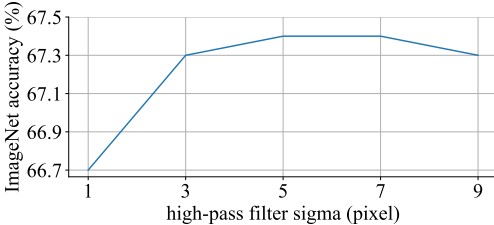

Figure 6: Impact of high-pass filter parameter $\sigma$ during pretraining on ImageNet-1K linear probe accuracy.

**View Sharing** - In our masking strategy, we apply the standard augmentations to generate multiple views and then randomly apply masks on these views. One alternative is to apply random masks on the same augmented view. Figure 5 shows that applying masks on the same view results in significantly worse representations.

As discussed in our design principle, the masks are used to prevent superficial solutions based on masked areas. It is still important to apply different augmentations to the original image to prevent the superficial solutions based on unmasked areas.

Table 5: Impact of view-sharing during pretraining on ImageNet-1K linear probe accuracy (in %). A view is generated with standard augmentations, including RandomResizedCrop, ColorJitter, HorizontalFlip, GaussianBlur, and Grayscale. Our standard approach applies masks to two different views. Here, we find that applying masks on a shared view results in significantly worse performance.

| Augmentation strategy | Accuracy |
|---|---|
| Same view + random mask | 29.1 |
| Different views + random mask | 65.6 |

# C    ADDITIONAL VISUALIZATION

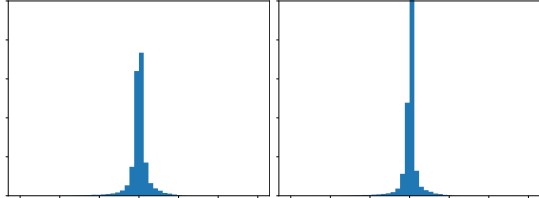

Figure 7: Color histograms of (left) high-pass filtered image and (right) high-passed image with mask.

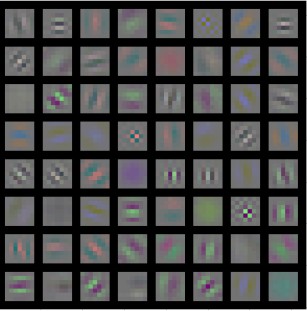

Figure 8: Visualization of convolutional kernels trained with MSCN

