# OpenReview forum: "Masked Siamese ConvNets: Towards an Effective Masking Strategy for General-purpose Siamese Networks "
_ICLR.cc/2023/Conference — Submitted to ICLR 2023_

### Official Review · Reviewer_Kw3b · 2022-10-22

**Confidence:** 5
**Correctness:** 3
**Technical Novelty And Significance:** 3
**Empirical Novelty And Significance:** Not applicable
**Recommendation:** 5

**Clarity, Quality, Novelty And Reproducibility:**

Overall the paper lacks some clarity in the explanations, and also the results are not clearly presented.

More specifically:
- What is the effect of the high-pass filter on the performance with non-masked images?
- There may be other ways of masking without introducing spurious edges.
- Figure 4 is not clear. What do you mean by "off-the-shelf ConvNet encoder"? The joint embedding loss is not detailed. Some details are missing on how the different (positive/negative) pairs are formed (within the batches).
- The relation to Assran et al. (2022) and Baevski et al. (2022) is not clear. Can MSCN be applied to ViT architectures (and be compared to MSN)?
- The text in section 6.1. states that the approach performs slightly worse for BYOL but from Table 1 this seems not to be the case.
- For Table 2, what is the task and dataset?
- It is not obvious how to interpret and read Table 4, and what to conclude. The underlining is somewhat arbitrary and confusing.
- The standard deviations are not presented in the results tables. I presume the results correspond to only one run.



**Strength And Weaknesses:**

Strengths:
- Good experimental results
- Genericity of the approach

Weaknesses:
- The description of the approach is unclear at some places
- Some of the results are not clearly presented



**Summary Of The Paper:**

This paper presents a general masking strategy for visual representation learning with siamese neural network architectures. The approach is applicable to CNN-based or ViT-based neural networks.
It consists in pre-processing the input with a high-pass filter and then doing different augmentations that mask out parts of the input images with a certain probability, inspired from the natural language processing domain.
Experimental results have been applied to different baseline models, notably to SimCLR and BYOL, and show an improved performance using the pre-trained models for image classification on ImageNet, and transfer lerning for classification, detection and instance segmentation on other datasets.


**Summary Of The Review:**

The presented method is interesting and relatively novel although largely inspired from other papers using masking. The results show the benefit of the proposed approach for self-supervised representation learning with general siamese neural network architectures. But the presented results lack some detail and rigor.

---

> ### Author Response · Authors · 2022-11-14
> **Reply to Reviewer Kw3b**
>
> We thank the reviewer for the insightful comments and suggestions. We have revised the manuscript and fixed the clarity issue. Here, we respond to each specific point raised by the reviewer:
>
> >What is the effect of the high-pass filter on the performance with non-masked images?
>
> Only adding a high-pass filter on the input has a negligible effect on the performance both in the supervised and self-supervised settings. Please refer to the "General Responses" above on "preprocessing and high-pass filtering".
>
> >There may be other ways of masking without introducing spurious edges.
>
> One other solution might be masking based on semantic segmentation, which has been explored in the paper"Characterizing and Improving the Robustness of Self-Supervised Learning through Background Augmentations".
> However, this type of masking requires even stronger prior knowledge and is hard to generalize to new domains.
>
> >Figure 4 is not clear. What do you mean by "off-the-shelf ConvNet encoder"? The joint embedding loss is not detailed. Some details are missing on how the different (positive/negative) pairs are formed (within the batches).
>
> Thanks for pointing this out.
> 1. "off-the-shelf ConvNet" means that our method works for any type of ConvNet backbone. We admit that this is unclear and may be misunderstood as pretrained. We have fixed it in our paper.
> 2. For the joint embedding loss, we wanted to show that our methods can directly work with various joint-embedding loss functions including BYOL, SimCLR, etc. Different loss functions have different behaviors in terms of positive/negative pairs. For each method, we closely follow the setting proposed in the original paper.
>
> >The relation to Assran et al. (2022) and Baevski et al. (2022) is not clear. Can MSCN be applied to ViT architectures (and be compared to MSN)?
>
> Please refer to general response #3 above on "Question Related to ViT Results".
>
> >The text in section 6.1. states that the approach performs slightly worse for BYOL but from Table 1 this seems not to be the case.
>
> We are sorry for the misunderstanding. The text in Sec 6.1 was trying to show that the improvement for BYOL is less significant than the improvement of SimCLR. We have changed our wording and made it more clear in the updated paper.
>
> >For Table 2, what is the task and dataset?
>
> In Table 2, the task is linear probing. The dataset is the ImageNet.
> Thanks for pointing this out. We have clarified the task and the dataset in the updated paper.
>
> >It is not obvious how to interpret and read Table 4, and what to conclude. The underlining is somewhat arbitrary and confusing.
> In general, MSCN method improves the objection detection results to be better or close to the SOTA.
>
> The underlined means 0.3 within the best result, following common practice.
>
> >The standard deviations are not presented in the results tables. I presume the results correspond to only one run.
>
> Thanks for pointing this out. For most of the experiments, our multiple runs have a standard deviation of accuracy within 0.1%. We follow the common practice of reporting accuracy at 0.1% precision to be consistent with the results from other papers. Following common practice, we will add more digits with standard deviation when releasing the code.

---

> ### Author Response · Authors · 2022-11-18
> **Has our response addressed your concerns?**
>
> Dear Reviewer Kw3b:
>
> We would be grateful if you kindly confirm whether our response has addressed your concerns and let us know if any issues remain. In the following, we recap the key points of our response:
> 1. We fixed the unclear phrases as mentioned in the original review, including the "off-the-shelf ConvNet", the task and dataset of table 2 and the wording for the performance regarding BYOL.
> 2. We answered the question regarding preprocessing and masking strategy.
> 3. We restated the motivation of our masking strategy and discussed potential other masking strategies that don't introduce spurious edges.
> 4. We clarified the joint embedding loss we used, the connection between our method and the methods using ViT, and responded to your concern on the standard deviations.
>
> We appreciate your constructive comments, which allowed us to improve our paper. We look forward to your feedback!

---

### Official Review · Reviewer_29aC · 2022-10-23

**Confidence:** 2
**Correctness:** 3
**Technical Novelty And Significance:** 2
**Empirical Novelty And Significance:** 3
**Recommendation:** 6

**Clarity, Quality, Novelty And Reproducibility:**

The clarity is good due to the simplicity of the proposed method and tricks. Also, it should be easy to replicate the results. Regarding the novelty, the reviewer is unsure if all the proposed components meet the bar of ICLR and need to discuss with other reviewers and the AC.

**Strength And Weaknesses:**

This paper makes the masking strategies available for convolutional-based siamese networks for SSL. The color histogram and convolutional visualization observation about the masking augmentation causing missing information and parasitic edges are interesting. The systematic studying of masking strategies can offer a way for other researchers to follow. Overall the paper is very easy to follow.

The experiments are conducted thoroughly, including various tasks (semi-supervised classification, transferred image classification, object detection, and instance segmentation) and datasets (IN-1K, iNaturalist, Places-205, VOC, and COCO). Results show the proposed method helps the standard SSL methods achieve higher performance. The related ablation studies are included to prove the effectiveness of different masking strategies.

The proposed preprocessing method may reduce the information in the image, which will cause the performance drop. In the current manuscript, all the comparisons include both the preprocessing methods and the comprehensive masking strategies; the reviewer is not clear if the preprocessing method brings improvements or masking strategies bring improvements. It would be better to test SimCLR, SimCLR + MSCN, BYOL, and BYOL + MSCN performance with simple/standard masking strategies.

How does the proposed processing method change the color histogram and learned convolution kernels, as shown in Figure 1? Since the proposed method is motivated by those observations, it would be more self-contained to include the updates once introducing the new methods.

**Summary Of The Paper:**

This paper proposed a pre-processing method to enable the masking strategy available for siamese networks with ConvNet backbone under self-supervised settings. It is motivated by performing random masking would cause missing information and parasitic edge issues. Also, systematic masking strategies are studied, including noisy mask, channel-independent mask (RGB), focal mask, and asymmetric augmentation. Extensive experiments on various tasks are conducted, including semi-supervised classification in IN-1K, transferred image classification results on iNaturalist and Places-205, and object detection & instance segmentation on VOC & COCO. The results show consistent improvements in combining the proposed processing method, high-pass filter, and masking strategies.

**Summary Of The Review:**

The reviewer feels that all the empirical studies presented in this paper will likely benefit the relevant community. However, the proposed masking strategies still heavily involve human priors. Also, the reviewer is unsure if the improvements come from introducing the preprocessing method or the hybrid masking strategies.

---

> ### Author Response · Authors · 2022-11-14
> **Reply to Reviewer 29aC**
>
> Thanks for your insightful and positive reviews, and we appreciate the valuable suggestions for the additional visualization! We have revised the manuscript. And the followings are the responses for each asked question:
>
> >The proposed preprocessing method may reduce the information in the image, which will cause the performance drop. In the current manuscript, all the comparisons include both the preprocessing methods and the comprehensive masking strategies; the reviewer is not clear if the preprocessing method brings improvements or masking strategies bring improvements.
>
> We showed in the paper that the preprocessing method will not cause the performance drop in the supervised setting.
> As for the question that whether the preprocessing method brings improvements or masking strategies bring improvements, please refer to general response #1 above on preprocessing and high-pass filtering.
>
> >How does the proposed processing method change the color histogram and learned convolution kernels, as shown in Figure 1? Since the proposed method is motivated by those observations, it would be more self-contained to include the updates once introducing the new methods.
>
> Thanks for the suggestion. We have added suggested information to our paper. Please kindly check the Appendix C

---

> > ### Comment · Reviewer_29aC · 2022-11-15
> > **Reviewer Response**
> >
> > Hi Authors,
> >
> > I thank you for your rebuttal.
> >
> > Regarding the new visualization, I wonder what the difference is between Figure 8 and Figure 1c? Also, according to Figure 7, this distribution is super extreme. Your point here is the distributions of high-pass filtered image and high-pass filtered image with mask are similar? Am I right?

---

> > > ### Author Response · Authors · 2022-11-16
> > > **Response to Reviewer 29aC**
> > >
> > > We thank the reviewer for the quick response.
> > >
> > > Figure 1c shows the learned convolutional kernels trained with (left) standard augmentation; (right) naive mask-only augmentation.
> > > Figure 8 shows the learned convolutional kernels trained with our proposed framework, i.e., preprocessing + our masking strategy.
> > > Figure 8 demonstrates similar edge-detecting behavior as Figure 1c (left), which proves the effectiveness of our proposed method.
> > >
> > > Figure 7 shows a relatively more extreme distribution because the high pass filter moves most of the value close to zero, which represents low-information pixels. Even though such extreme distribution indicates that there is missing information in the input, our experiments show that it won't harm the learned representations.
> > >
> > > Yes, similar distributions prove that with a highpass filter, adding masks won't create a superficial solution of naively using the color histogram as the feature.
> > >
> > > Please let us know if we have successfully addressed your question!

---

> > > > ### Comment · Reviewer_29aC · 2022-11-24
> > > > **Reviewer Response**
> > > >
> > > > I thank the efforts made by the authors.
> > > >
> > > > I feel the current explanations for Figure 7 are quite intuitive, with only empirical support. The extremely high frequency made by introducing masks may still bring issues.
> > > >
> > > > Overall the paper provides many empirical studies for the community, which may be worthy of presenting at ICLR. I am not confident about this.

---

> ### Author Response · Authors · 2022-11-18
> **Has our response addressed your concerns?**
>
> Dear Reviewer 29aC:
>
> Would you mind kindly confirm whether our response has addressed all your concerns? Please let us know if you have any other comments. We are appreciate any feedback from you.

---

### Official Review · Reviewer_nzgE · 2022-10-24

**Confidence:** 5
**Correctness:** 3
**Technical Novelty And Significance:** 3
**Empirical Novelty And Significance:** 3
**Recommendation:** 5

**Clarity, Quality, Novelty And Reproducibility:**


The quality and clarity are median. The originality is good.


**Details Of Ethics Concerns:**


There is no ethics concern.


**Strength And Weaknesses:**


(Positive) The subject studied in this paper is important. Exploring the role of masking as augmentation in self-supervised learning of convolutional networks is encouraged.

(Positive) After analysis, the authors believe that convolutional networks cannot correctly ignore the null information of masked areas through the attention mechanism like VIT. This is very reasonable.


(Positive) The authors argue that masking brings artificial edges, which will hurt the learning of convolutional neural networks. This is very reasonable.


(Negative) When the authors discuss Siamese networks, their formulation ignores the commonly used and very important projectors in SSL algorithms. Please add a projector to the formulation.

(Positive) Regarding the principles that augmentation needs to adhere to in self-supervised learning, the author's conclusion is correct: Preventing easy solutions is very important; the augmentations are hoped to be class-aware.


(Positive) The authors use high-pass filtering to suppress the null information in the masked area, which is very clever.


(Positive) The authors use high-pass filtering to suppress artificial edges, which is also very clever.


(Negative) However, in the masking strategy, the author uses a noisy mask. I think this contradicts the motivation of the authors. We all know that noise masks, especially Gaussian noise, bring a lot of high-frequency signals. High-pass filtering preserves these signals. The authors claim that this kind of noisy mask is useful, isn't that slapping one's own face?


(Positive) The Channel-Independent Mask proposed by the authors is a bit interesting.

(Negative) The caption "Few-shot Semi-supervised Learning on ImageNet-1K" in Table 1 is misleading. Also included here is an evaluation of the very important linear probing in SSL. Please modify this caption.


(Negative) The proposed method has obvious gains on SimCLR. But the gain is not obvious on BYOL.

(Negative) The application of the proposed method in ViT even leads to performance degradation, which is very worrying.

(Negative) In the Object detection and instance segmentation tasks, the authors are missing one of the most important comparison objects, which is the baseline on which the authors' method is based. I'm guessing it could be BYOL or SimCLR. Given that the authors' method doesn't gain much, I'm worried that the authors' method won't outperform this baseline.


(Negative) The authors have repeatedly emphasized that masking is the most general augmentation. This is worrying. This expression is too subjective.


(Negative) In fact, the biggest challenge of Masking augmentation in convolutional networks is masked image modeling. Unfortunately, after reading the whole article, I found that the authors' method does not solve this problem. This is quite disappointing for readers.




**Summary Of The Paper:**


Summary:
This paper explores solving the masking augmentation problem in convolutional neural networks. This topic is very important. The authors' method is also very clever. However, unfortunately, its performance is not satisfactory. Moreover, it does not solve the more important problem of masking image modeling. Therefore, this article is a borderline article, leaning towards being rejected. If the authors can answer my important questions, I will consider raising the rating. If this article can solve the problem of masking image modeling, I will give it a high score.


**Summary Of The Review:**


See "Summary Of The Paper." This article is a borderline article, leaning towards being rejected. If the authors can answer my important questions, I will consider raising the rating. If this article can solve the problem of masking image modeling, I will give it a high score.

---

> ### Author Response · Authors · 2022-11-14
> **Reply to Reviewer nzgE**
>
> We thank the reviewer for the valuable comments. We appreciate all the positive and negative points that allow us to improve our paper.
> We have revised the paper based on the suggestions. Please kindly find the reply to each listed negative point below:
>
> >When the authors discuss Siamese networks, their formulation ignores the commonly used and very important projectors in SSL algorithms. Please add a projector to the formulation.
>
> Thanks for the suggestion. Indeed, a projector is an important module in all siamese network frameworks. The discussion in Section 3 assume that projector is included in the encoder and we have clarified it in formulation. Please check the paragraph 1 of section 3.1
>
> >However, in the masking strategy, the author uses a noisy mask. I think this contradicts the motivation of the authors. We all know that noise masks, especially Gaussian noise, bring a lot of high-frequency signals. High-pass filtering preserves these signals. The authors claim that this kind of noisy mask is useful, isn't that slapping one's own face?
>
> Please refer to general response #1 above on preprocessing and high-pass filtering.
>
> >The caption "Few-shot Semi-supervised Learning on ImageNet-1K" in Table 1 is misleading. Also included here is an evaluation of the very important linear probing in SSL. Please modify this caption.
>
> Thanks for pointing this out. We have corrected the Table caption in our paper.
>
> >The proposed method has obvious gains on SimCLR. But the gain is not obvious on BYOL.
>
> Please refer to general response #2 above on performance regarding BYOL.
>
> >The application of the proposed method in ViT even leads to performance degradation, which is very worrying.
>
> We are sorry for the confusion. The ViT results presented in Table 2 are not from our model. The ViT + Mask is from the Masked Siamese Network paper.
> We put ViT results in Table 2 (Effect of Masking on ConvNets and ViTs) to show that ConvNet with our masking strategy and ViT with naive masking have similar performances when comparing to standard augmentation approaches.
> Please also refer to general response #3 above on Question Related to ViT Results.
>
> >In the Object detection and instance segmentation tasks, the authors are missing one of the most important comparison ob
> jects, which is the baseline on which the authors' method is based. I'm guessing it could be BYOL or SimCLR. Given that the authors' method doesn't gain much, I'm worried that the authors' method won't outperform this baseline.
>
> Thanks for pointing out the missing comparison, we have added those results in our paper.
> Our model actually outperforms the baselines (BYOL and SimCLR) on object detection tasks.
>
>
> >The authors have repeatedly emphasized that masking is the most general augmentation. This is worrying. This expression is too subjective.
>
> Thanks for pointing this out. We admit the statement that masking is the most general augmentation is an overclaim. We focus on masking because this is a known successful augmentation that can be applied to multiple modalities, including image, text, audio, etc. We have removed the over-repeated phrase in the paper.
>
> >In fact, the biggest challenge of Masking augmentation in convolutional networks is masked image modeling. Unfortunately, after reading the whole article, I found that the authors' method does not solve this problem. This is quite disappointing for readers.
>
> Thanks for bringing out this important question on whether our method can be applied to MIM. As mentioned in our discussion section (Masked ConvNets for Masked Image Models), there are still challenges to applying MIM on ConvNets. Here, we would like to share more thoughts:
> 1. There are shared problems of Masked ConvNets for Siamese Networks and Masked ConvNets for Masked Image Modeling. Both will suffer from parasitic edges and missing information. Therefore, we believe the preprocessing steps will benefit the MIM approach.
> 2. The masking strategy serves different purposes in siamese networks and MIM. For MIM, it only removes information for the model to predict. For siamese networks, the masking must explicitly guide the network to focus on the different type of features and avoid trivial ones. e.g. noise mask prevents trivial color histogram solution. Therefore, we believe some other design may be required for MIM to work well with masked ConvNets.
> 3. Solving MIM with ConvNets will be our future work.

---

> ### Author Response · Authors · 2022-11-18
> **Has our response addressed your concerns?**
>
> Dear Reviewer nzgE:
>
> We would be grateful if you can confirm whether our response has addressed your concerns and let us know if any issues remain. In the following, we recap the key points of our response:
> 1. We fixed the issues mentioned in the original review, including adding projector to the formulation, changing the caption of table 1 and softening the claim that masking is the most general augmentation.
> 2. We clarified the misunderstanding about the ViT results and noisy mask.
> 3. We answered the question about performance regarding BYOL.
> 4. We added extra object detection results to show our method boosts the object detection performance comparing to the original SimCLR and BYOL methods.
>
> We are looking forward to your feedback!

---

> ### Comment · Reviewer_nzgE · 2022-12-11
> **Responses to the authors' responses:**
>
>
> I really appreciate the great effort the authors have made to respond to my concerns. For example, the authors successfully addressed my concerns regarding the rigor of the article description, the completeness of the table description, the clarification of confusion, and the objectivity of the article description.
>
> It is especially commendable that the authors report the baseline of the detection task and show that their method is effective in the detection task.
>
> However, I still feel that the shortcomings of this paper are obvious. Regarding Gaussian noise, I have always recognized the effectiveness of Gaussian noise. But the example of Gaussian noise shows that the effectiveness of this paper may be explained more than high-pass filtering. Hence, the authors almost had to rewrite the whole paper. If the authors can give a better explanation and motivation and rewrite the paper, I think it will be a better article. Also, I am very grateful to the authors for their discussion of MIM. When I first read this article, I thought the authors would solve the MIM problem, and I was thrilled. But after reading it, I was a little disappointed to find that the authors didn't address this issue.
>
>
> To sum up, the method proposed by the article is worthy of recognition, but the example of Gaussian noise is incompatible with other examples, which shows that understanding the method of this article needs more perspectives. I hope the authors can improve the paper and resubmit it. If I were still a reviewer of the new submission, I would raise my rating. If the authors solve the MIM problem, it will be a very influential work. I am expecting improvements from the authors.

---

### Author Response · Authors · 2022-11-14
**General Response to All Reviewers**

First of all, we would like to thank all reviewers for their constructive comments, which allowed us to improve our paper. We appreciate that all reviewers find our approach novel and agree on the importance of the problem that we are trying to solve.  In the general reply, we address a few common questions and present some additional results. We also specifically respond to each reviewer for more specific questions.

### 1. Preprocessing and High-Pass Filtering

Both reviewer 29ac and reviewer Kw3b raise the concern that whether the preprocessing method itself is the reason for the performance improvement instead of the masking strategy. Based on our experiments, applying the high-pass filter only doesn't change the performance much. The high-pass filter only boosts the performance when combined with our masking strategy.

Reviewer nzgE raises the concern over the noisy mask. The high-pass filter only preserves the high-frequency component of the input signal, and the noisy mask introduces some high-frequency noise to the input signals. We believe that the noisy mask can be considered a form of data augmentation. Since the two crops have different high-frequency noises (we use a different noisy mask for each crop), the encoder has to learn to denoise the signal and capture the mutual information from the high-frequency component preserved by the high-pass filter.

We conducted additional experiments showing that the noisy mask improves the linear probe performance from 40.0% to 48.2% with Resnet50 backbone and SimCLR loss (100ep pertaining). We will add the corresponding experiments in the final paper.

### 2. Performance Regarding BYOL

Both the reviewers nzgE and reviewer Kw3b questioned the performance gain of BYOL (+MSCN). Our method improves BYOL linear probe and also significantly improves its few-shot classification performances.
Its performance gain is less significant compared to those on SimCLR. This is mainly because BYOL itself depends less on the augmentation, as shown in BYOL paper Figure 3, while our method essentially improves the augmentation.
In addition, as we mentioned in the paper, one of our main empirical contributions is showing that adding masks to siamese networks improves the performance of few-shot settings without affecting the performance of linear probing. This result is consistent with the Masked Siamese Networks. Together with the MSN paper, we show that improving few-shot settings is an empirical advantage for siamese networks with masking.

### 3. Question Related to ViT Results

> Misunderstanding on Table 2

We are sorry for the misunderstanding on the ViT experiments in Table 2. These ViT results are not from our experiments but from the reference paper. They are used to show that the Masked Siamese Networks with ConvNets with our method is as good as that with ViT (By MSN paper). We have made this clear in the caption and corresponding section in the paper.

> Can our method be applied to ViT?

The reviewer asked whether our method can be applied to ViT backbones. We want to clarify that our motivation is to allow ConvNet to benefit from the masked siamese framework, in the same way as ViT, by solving the introduced problems on the masked input, such as parasitic edges. Thus, ViT backbones can easily avoid such issues by using the masking strategy proposed in MSN/MAE. Therefore, for ViT backbones, it will be more appropriate to use Masked Siamese Network, and it is not necessary to apply the extra preprocessing proposed in our paper. We will clarify this in the final paper.

---

### Decision · Program_Chairs · 2023-01-20

**Decision:**

Reject

**Justification For Why Not Higher Score:**

There are several unaddressed issues in the reviews, such as the experimental results being not very satisfactory, and the paper writing should be further improved.

**Justification For Why Not Lower Score:**

N/A

**Metareview: Summary, Strengths And Weaknesses:**

The paper explores different masking augmentation strategies for self-supervised learning. The reviewers have mixed opinions on the paper, with some praising its importance and clever methods, but others noting that its performance is not satisfactory. The reviewers also raise concerns about the quality and clarity of the paper. Overall, the paper needs to be further improved and incorporated review feedback in order to meet the acceptance bar of ICLR.